# Chloroplast Genome Draft of *Dryobalanops aromatica* Generated Using Oxford Nanopore Technology and Its Potential Application for Phylogenetic Study

**Dwi Wahyuni** [1]**, Fifi Gus Dwiyanti** [1,2]**, Rahadian Pratama** [2,3]**, Muhammad Majiidu** [2]**, Henti Hendalastuti Rachmat** [4] **and Iskandar Zulkarnaen Siregar** [1,2,*]

1   Department of Silviculture, Faculty of Forestry and Environment, IPB University, Bogor 16680, Indonesia; dwi_wahyuni@apps.ipb.ac.id (D.W.); fifi_dwiyanti@apps.ipb.ac.id (F.G.D.)
2   Advanced Research Laboratory (ARLab), IPB University, Bogor 16680, Indonesia; rahadian@apps.ipb.ac.id (R.P.); mmajiidu@gmail.com (M.M.)
3   Department of Biochemistry, Faculty of Mathematics and Natural Science, IPB University, Bogor 16680, Indonesia
4   Forest Research and Development Center, Ministry of Environment and Forestry, Gunung Batu, Bogor 16118, Indonesia; hendalastuti@yahoo.co.uk
*   Correspondence: siregar@apps.ipb.ac.id

**Abstract:** Kapur (*Dryobalanops aromatica*) is an important dipterocarp species currently classified as vulnerable by the IUCN Red List Threatened Species. Science-based conservation and restoration efforts are needed, which can be supported by new genomic data generated from new technologies, including MinION Oxford Nanopore Technology (ONT). ONT allows affordable long-read DNA sequencing, but this technology is still rarely applied to native Indonesian forest trees. Therefore, this study aimed to generate whole genome datasets through ONT and use part of these data to construct the draft of the chloroplast genome and analyze the universal DNA barcode-based genetic relationships for *D. aromatica*. The method included DNA isolation, library preparation, sequencing, bioinformatics analysis, and phylogenetic tree construction. Results showed that the DNA sequencing of *D. aromatica* resulted in 1.55 Gb of long-read DNA sequences from which a partial chloroplast genome (148,856 bp) was successfully constructed. The genetic relationship was analyzed using two selected DNA barcodes (*rbc*L and *mat*K), and its combination showed that species of the genus *Dryobalanops* had a close relationship as indicated by adjacent branches between species. The phylogenetic tree of *matK* and the combination of the *mat*K and *rbc*L genes showed that *D. aromatica* was closely related to *Dryobalanops rappa*, whereas the *rbc*L gene showed group separation between *D. aromatica* and *D. rappa*. Therefore, a combination of the *mat*K and *rbc*L genes is recommended for future use in the phylogenetic or phylogenomic analysis of *D. aromatica*.

**Keywords:** *Dryobalanops aromatica*; genome; markers; MinION; sequencing



## 1. Introduction

Dipterocarpaceae is an ecologically and economically important tree family widely used for timber and non-timber forest products. The non-timber forest products of Dipterocarpaceae, such as resin, sap, and camphor, have high economical values [1]. *Dryobalanops aromatica*, an important species from this family that produces camphor, is native to Indonesia and is distributed in North Sumatra, Riau Islands, and Borneo (West Kalimantan, Sabah, and Sawarak) [2]. Biologically, *D. aromatica* tree can reach a height of 65 m with a clear-bole height of 30–40 m and an average diameter at breast height of 70 cm when fully mature [3,4], living at an altitude of <500 m above sea level [5]. The camphor of *D. aromatica* contains a borneol compound [6] that can be utilized in the form of crystals and resins as a material for preparing perfumes and various medicine [7]. In China, it is used as an antiseptic, anti-inflammatory, analgesic, and as an additive for sanitary napkins

(bio panty) to reduce pain during menstruation [8]. A recent study has found that borneol in *D. aromatica* can be used in diseases of the nervous system [9]. Borneol can dilute the blood in the human brain blood vessels and exert cerebral protective effects. A review [10] concluded that borneol has potential in preventing the enlargement of infarction, making it a neuroprotective agent against cerebral ischemia. At present, populations of *D. aromatica* are declining because the crystals obtained from the stems of *D. aromatica* have a higher market price than oil, leading to uncontrolled harvesting methods [11]. As a result, Kapur (*D. aromatica*) has been categorized as vulnerable in the 2018 IUCN (International Union for Conservation of Nature and Natural Resources) Red List of Threatened Species [12]. Therefore, a concerted effort is needed to increase the population of *D. aromatica* through tree regeneration, which could be aligned with ecosystem restoration programs promoted by the United Nations [13].

Tree regeneration and genetic resource conservation experiments require a science-based genetic conservation strategy to provide complete genomic data and information on target species [14,15]. Genomic tools are indispensable because the current molecular methods for forest trees still have DNA fragments and limited DNA sequence analysis. Nowadays, DNA sequencing technologies rapidly evolve and have entered third-generation sequencing through long-read sequencing analysis. Recent technological developments can support the sequencing process with portable and economic devices, such as MinION sequencer from Oxford nanopore Technologies (ONT). This nanopore technology has the advantage of being able to sequence large eukaryotic genomes and minimize the presence of fragments that cause gaps between DNA strand assemblies [16]. This technology will also aid in chloroplast genome research. The chloroplast genome is characterized by having a circular structure and containing a pair of long inverted repeats that can add several errors to sequencing results by using short-read sequencing technologies [17]. Meanwhile, long-read sequencing can capture entire structural variants or repetitive segments in a single DNA read [18]. The utilization of long-read DNA sequencing technology will benefit the construction of the plant genome, which consists of repetitive DNA sequences and single-copy gene sequences.

With the advent of ONT, many native tropical trees, especially those with high economic potentials such as *D. aromatica*, can be further investigated as a start-up genomic research activity to support conservation, breeding, and ecosystem restoration by informing seed sourcing, species selection, or provenance selection [19]. In addition, this portable sequencing enables in situ sequencing used in various field conditions, including real-time sequencing of endangered species, which can help democratize scientific research and conservation efforts [20]. The data resulting from this portable sequencer will generate important genetic information, such as analyses of genetic relationships that can be used as baseline data in determining the conservation strategy for the species and providing new insights for evolutionary studies. Analysis of genetic relationship can be performed with the chloroplast genome. The chloroplast genome is the perfect source for phylogenetic study because of its stable genome structure, higher evolutionary rate than the mitochondrial genome, and 100–130 genes that encode ~79 proteins, ~30 transfer RNAs, and 4 ribosomal RNAs [21]. In the present study, we used universal gene markers *mat*K and *rbc*L to construct phylogenetic trees because these two genes are considered standard plant DNA barcoding markers with high discriminatory power between angiosperms [22,23]. This study aimed to generate whole genome datasets and use part of them as a promising showcase to construct the draft of the chloroplast genome and analyze the universal DNA barcode-based genetic relationships for *D. aromatica*.

## 2. Materials and Methods

### 2.1. Plant Materials

Samples of one silica-gel dried twig and one fresh leaf derived from one individual healthy *D. aromatica* seedling (Figure 1) were collected for later laboratory work. Silica-gel dried twigs were used because they allow easier collection of sawdust with a drill tool than

fresh twigs. The seedling was originated from Lingga Island in Riau Archipelago and has been raised for 5 years in the Komatsu-FORDA Conservation nursery, Forest Research and Development Center, Forestry and Environmental Research Development and Innovation Agency, Ministry of Environment and Forestry in Bogor, West Java, Indonesia.

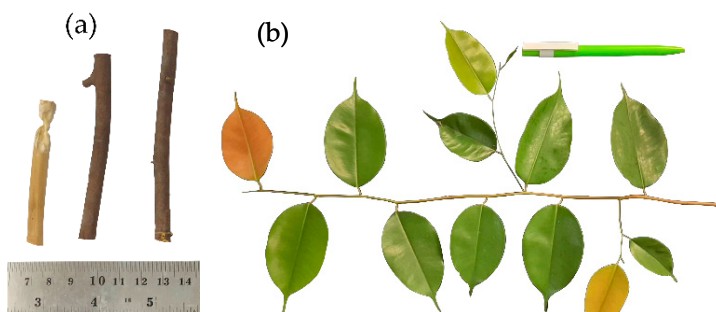

**Figure 1.** *Dryobalanops aromatica* samples were used in this study. Silica-dried twig (**a**); Fresh leaves (**b**).

## 2.2. Genomic DNA Extraction

Genomic DNA was extracted by using the modified cetyl trimethyl ammonium bromide (CTAB) method [24]. The CTAB buffer contains 1% PVP, 5 M NaCl, 0.5 M EDTA, 10% CTAB solution, and $dH_2O$. *D. aromatica* fresh leaf was mixed with CTAB buffer and ground manually using a mortar and pestle before proceeding to the extraction process. Meanwhile, the twig was drilled to obtain fine sawdust using a Dremel 3000 Rotary Tool before being mixed with CTAB buffer and mashed with a mortar and pestle for further DNA extraction. The quality of genomic DNA was evaluated using agarose gel electrophoresis performed by Mupid exU, and the purity of genomic DNA was assessed using Nanophotometer IMPLEN NP80. The A260/280 ratio of 1.8 is recommended for sequencing. DNA quantity was measured with a Qubit 1.0 fluorometer using the Qubit dsDNA BR assay kit. The quantity of DNA required for sequencing was 1 µg of DNA in 48 µL volume. The use of this Qubit is recommended because it provides accurate measurements that can distinguish DNA from the rest of the solute [25].

## 2.3. DNA Library Preparation and Sequencing

A DNA library was prepared in accordance with the nanopore protocol using Native Barcoding Genomic DNA with barcode kit 1-12 (EXPNBD104) and ligation sequencing kit (SQK-LSK109), version NBE_9065_v109_revJ_23May2018. The DNA library preparation consists of several steps: (i) DNA repair (FFPE) and end-prep for optimizing DNA quality, (ii) preparing DNA sequence ends for barcode and adapter attachment, and (iii) preparing R9.4 flow cell for sequencing (priming the flow cell). DNA clean-up was performed in between each library preparation step using magnetic AMPure XP beads. The DNA library was loaded into the R9.4 MinION Flowcell, and sequencing was performed using MinKnow software from ONT.

## 2.4. Data Analysis

### 2.4.1. Sequence Raw Data Analysis

The output of MinION sequencing was raw Fast5 data, which was subsequently base called into FASTQ files using the Guppy program v4.2.3+8aca2af8 [26,27]. Then, a data quality check was performed using the NanoStat program v1.5.0 [28] to obtain the statistic of the FASTQ reads and its distribution of quality scores. The NanoPlot program v1.33.1 [29] was used to create a plot of reading length x average quality score. Reads with inadequate quality (Q < 7) and length <500 bp were filtered using the NanoFilt program v2.7.1 [30], and the parameters used were *-l 500 -q 7 –headcrop 10 –tailcrop 10 –readtype 1D*. The QC-passed reads were assembled using Rebaler (v0.2.0) [31], with *Dipterocarpus turbinatus* (NCBI accession code NC_046842.1) as the reference-based assembly [32], for correction of reads and assembled reads into contigs. The resulting contigs were subject to

assembly polishing using the MEDAKA consensus v1.2.1 [33] to obtain contigs with high accuracy. The statistics of the polished contig were calculated using QUAST v5.0.2 [34] with reference to the *D. turbinatus* chloroplast genome. Subsequently, the polished contig was annotated using the GeSeq platform [35], and GenBank annotations were generated.

2.4.2. Chloroplast Marker Analysis

The GenBank annotation obtained from GeSeq was visualized using SnapGene v5.2.3 [36] to search for the potential gene marker. Two universal coding gene sequences (Table S1), *rbc*L and *mat*K [22,23], were then selected for further analysis. Sequences of the *rbc*L and *mat*K genes were blasted on the NCBI [37] by using translated nucleotide query (BLASTX) [38–40] under the Dipterocarpaceae family. The result of 50 homolog sequences of each marker was downloaded in the form of a Fasta file for further phylogenetic tree construction.

2.4.3. Phylogenetic Tree Construction

Phylogenetic analysis was performed using MEGA X v10.2.2 [38,41]. Sequence alignment of 50 homolog sequences plus marker was carried out using ClustalW alignment and default parameter. A phylogenetic tree was constructed on the aligned sequences using the neighbor-joining algorithm and a bootstrap value of 1000 repetitions to test the topological validity of the phylogenetic tree [40]. The constructed tree was evaluated, and branches with bootstrap value >70% were retained. According to [42], the bootstrap value was categorized into very weak (<50%), weak (50–69%), moderate (70–85%), and high (>85%). Therefore, the bootstrap value must be at least >70% to obtain a topology with the reliable (valid) genetic relationship of *D. aromatica*. The final constructed phylogenetic tree was exported to Newick format (.nwk) and then uploaded on the iTOL web server [43] to create a phylogenetic tree cladogram design. The phylogenetic tree cladogram was finalized in Inkscape v1.0.2 [44] to provide a clear branch color thickness.

**3. Results**

*3.1. Genome Sequencing and Assembly*

The first step in the long-read analysis is base-calling or conversion from raw data to nucleic acid sequences. The MinION platform outputs in the form of FAST5 files, which are then converted into FASTQ (raw data from base-calling) [27]. The FASTQ files were subject to a quality check to determine the read length with its initial quality. On the basis of the distribution (Figure 2), the longest read lengths reach approximately 60 Kb or 60,000 bp with the highest reading quality of Q25 and the lowest quality of Q4. The higher the read lengths, the lower the number of reads. Most of the reads fall under 20 Kb and quality above Q10. Thus, the sequence of *D. aromatica* obtained in this study is good for long-read sequencing.

FASTQ data were filtered to remove sequences whose DNA quality is <Q7 according to the ONT quality passing standard [45]. DNA sequences with read lengths below 500 bp were removed to avoid wasting computational resources in the assembly process [46]. Previously, the results of the initial data quality examination showed that the genomic data of *D. aromatica* still had several base sequences that could increase or affect the error value due to low read length and quality. When low read length and quality were removed, the mean read length, mean read quality, and read length N50 statistically increased (Table 1). After filtering, approximately 96% of reads passed the quality control (351,411 reads) with a reading length N50 of 6114 bp and a total base of 1.55 Gb.

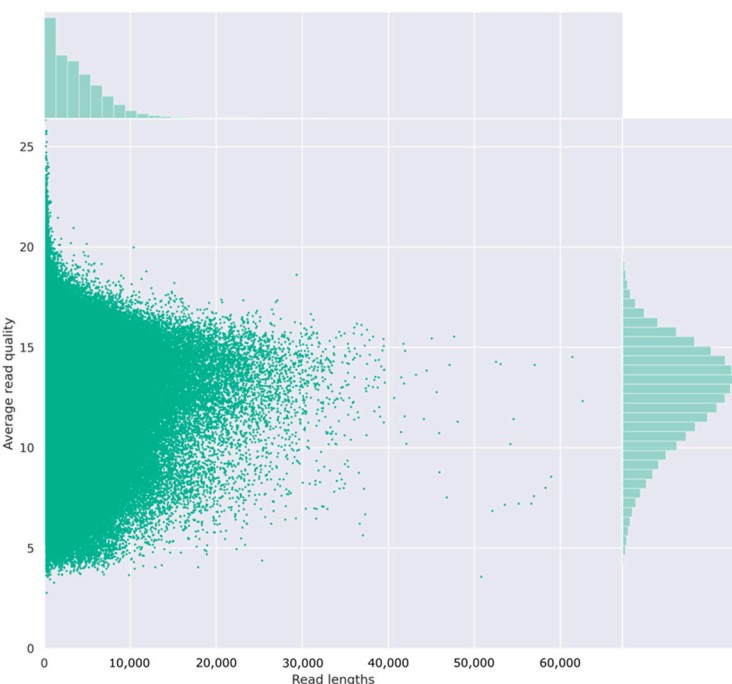

**Figure 2.** Histogram of read length distribution data and average read quality.

**Table 1.** Statistics of the raw, filtered, and assembled reads.

|  | Raw Reads | Filtered Reads | Assembled Reads |
|---|---|---|---|
| Mean read length/contig length (bp) | 3862 | 4438 | 148,856 |
| Mean read quality | 12,6 | 12,7 | - |
| Number of reads/contig | 418,943 | 351,411 | 1 |
| Read length N50 (bp) | 6061 | 6114 | - |
| Total bases (bp) | 1,617,953,241 | 1,559,878,347 | - |
| Average coverage | - | - | 186.804 |

The assembly stage in this study was carried out using reference-guided DNA assembly by comparing the studied genome with the reference genome in bioinformatics analysis. The reference-guided assembly produced a partial genome of *D. aromatica* chloroplasts of 148,856 bp. The GC content was calculated as 36.92%, which is consistent with cpDNAs from other Dipterocarpaceae family members, such as *Hopea reticulata* (37.4%) [47] and *Parashorea chinensis* (37.1%) [48]. Several genes with high GC content were exhibited by four ribosomal proteins, namely, *rrn23*, *rrn16*, *rrn4,5*, and *rrn5* with 55%, 56%, 50%, and 51%, respectively. In addition, the total genome fraction found in the partial genome was 89.99%, with 411 indels and 135,411 alignments for reference.

Reference assembly is less time-consuming and has computational power [49]. DNA assembly to generate the whole genome starts with combining overlapping reads to construct contigs. The contigs were combined to make scaffolds, which were also combined to obtain the whole genome. However, genome assembly usually meets several challenges (sequencing error, short reads, repeats, polymorphism, etc.) that need to be resolved and requires repeated sequencing before being able to construct a complete genome. Therefore, this study focused on the chloroplast genome of *D. aromatica* because of the single sequencing generated in this study.

### 3.2. Chloroplast Genome Annotation

Genome annotation was performed to identify functional genes along the genome sequence [50]. The annotation of *D. aromatica* chloroplast identifies genes contained in the

cpDNA (Table 2). The assembled draft chloroplast genome contains 137 genes, including 98 protein-coding sequences, 33 tRNA genes, and 8 rRNA genes. A total of 14 protein-coding genes contain intron, whereas 2 genes (*rps*12 and *ycf*3) contain 2 introns each. *trn*K-UUU exhibits the largest intron at 2528 bp, encompassing the *matK* gene. The smallest intron is located at the *rps*12 gene with an intron size of 232 bp (Table S2).

**Table 2.** Summary of the genome assembly and annotation in *D. aromatica*.

| Assembly Size (contig, bp) | 148,856 |
| --- | --- |
| GC content (%) | 36.92 |
| Genome fraction (%) | 89.99 |
| Number of indels | 411 |
| Alignment to reference | 135,411 |
| Total number of gene features | 349 |

The chloroplast genome draft contains genetic information in the form of protein-coding regions (Figure 3), which can be used as DNA barcode markers. Several cpDNA markers found in the draft genome of *D. aromatica* chloroplasts are common markers that have been developed and used as DNA barcodes; these markers include *rbc*L, *mat*K, the intergenic spacer *trn*H–*psb*A, and *trn*L-*trn*F [51]. The 98 protein-coding genes have been classified into genes responsible for photosynthesis (*psa*A, *psa*B, *psb*A, *psb*K, *pet*A, *pet*N, *atp*A, *atp*F, etc.), genes with self-replication (*rpl*33, *rpl*20, *rps*16, *rps*2, *rrn*16, *rrn*23, etc.), and other genes (*acc*D, *ccs*A, *cem*A, *inf* A, *mat*K, *rbc*L, etc.). On the basis of these findings, DNA barcodes *rbc*L and *mat*K and their combination were selected to be used in further analysis.

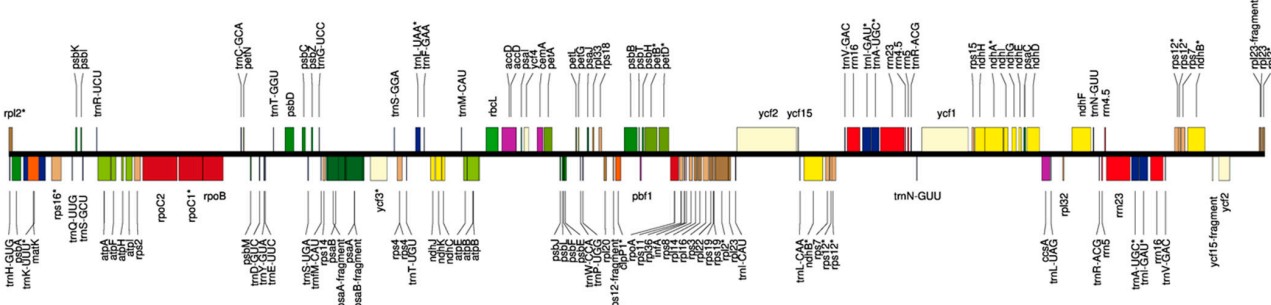

**Figure 3.** Partial assembly of the *D. aromatica* chloroplast genome.

### 3.3. Phylogenetic Tree Construction

The *rbc*L and *mat*K markers are commonly used as molecular identifications, especially plants for DNA barcoding in taxonomic studies. On the basis of the Consortium for the Barcode of Life's, the two markers (*rbc*L and *mat*K) and their combination were recommended for determining the genetic status of a species in a comprehensive, accurate, and fast manner [52,53]. The *rbc*L gene in cpDNA encodes RuBisCo and has a high degree of similarity between species because of its low mutation rate [54]. The *rbc*L gene has a full length of approximately ±1400 bp [55]. The *mat*K gene marker has a sequence length of ±1500 bp and has a gene substitution rate six times higher at the amino acid level and three times higher at the nucleotide level when compared with the *rbc*L gene because the *mat*K gene is the fastest evolving coding region that consistently shows high-level discrimination in species [52]. Previously, molecular phylogenetic studies have been carried out on the Dipterocarpaceae family using several markers [56,57], indicating the genetic relationship of a species observed in the phylogenetic tree. Phylogenetics is a genetic relationship diagram that can be analyzed based on the similarity of genes or phenotypes from one family; it is supported by several literature reviews as a complement to ecological and morphological information [58].

In this study, the results of genetic relationship analysis using the neighbor-joining method with 1000 bootstraps on *D. aromatica* with other species from the Dipterocarpaceae family showed that species of the genus *Dryobalanops* had a close relationship because they were in adjacent branches [59], either based on the *mat*K gene with a length of 1314 bp (Figure 4), the *rbc*L gene with a length of 1428 bp (Figure 5), or a combination of the two genes (Figure 6). However, the number and members of species in a branch were different for each gene used. For example, *Dryobalanops lanceolata* and *D. aromatica* were found in the phylogenetic tree based on the *rbc*L gene (Figure 4); *Dryobalanops beccarii*, *D. aromatica*, and *Dryobalanops rappa* were found in the phylogenetic tree based on the *mat*K gene (Figure 5); and *D. rappa* and *D. aromatica* were present in a phylogenetic tree based on the combined genes (Figure 6).

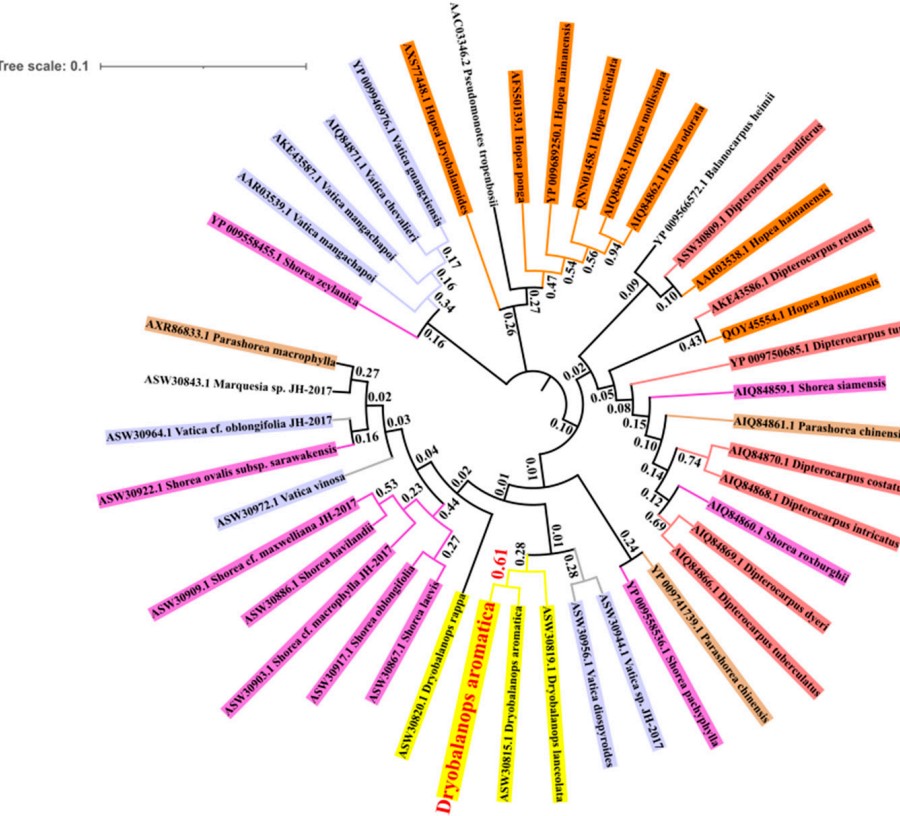

**Figure 4.** Phylogenetic tree of *D. aromatica* based on *rbc*L gene sequences.

In addition, the phylogenetic tree constructed from the *rbc*L gene showed that *D. aromatica* was genetically close to another *D. aromatica* sequence with a bootstrap value of 61%. However, *D. rappa* was outside the branch of the *Dryobalanops* genus and separated from *D. aromatica*. Meanwhile, the phylogenetic tree of *mat*K and the combination of *mat*K and *rbc*L genes showed a close relationship between *D. aromatica* and *D. rappa* with bootstrap values of 64% and 95%, respectively. The *rbc*L gene has a high success rate in amplifying gene fragments [60], but this gene also has some disadvantages, such as low resolution in distinguishing several closely related species [61]. Therefore, the *rbc*L gene was inconsistent in distinguishing between genera. By contrast, the *mat*K gene can better differentiate between genus and species levels [62].

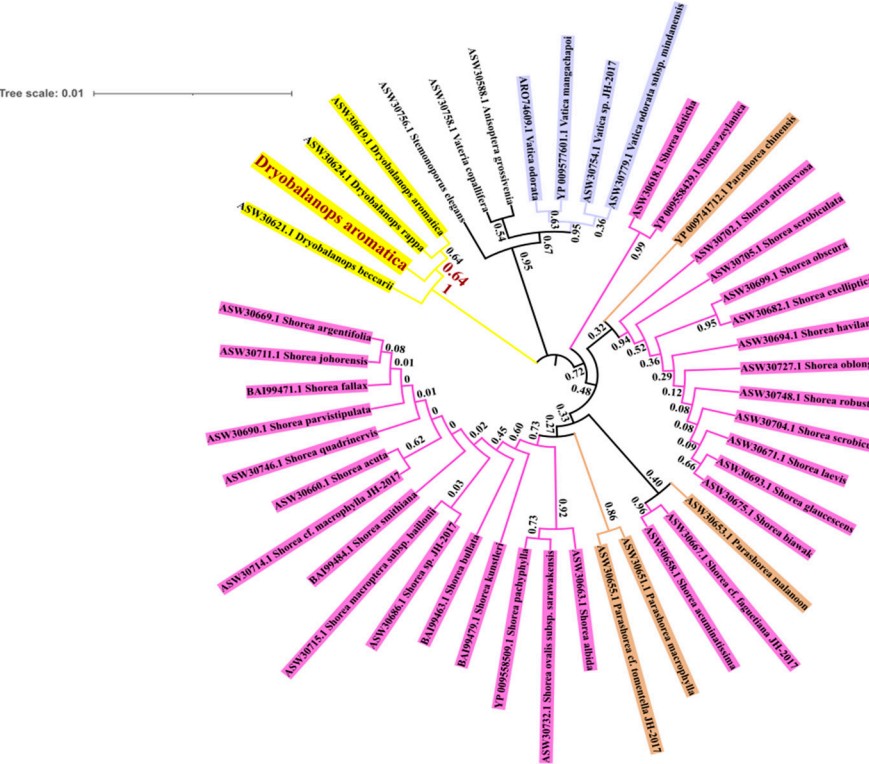

**Figure 5.** Phylogenetic tree of *D. aromatica* based on *mat*K gene sequences.

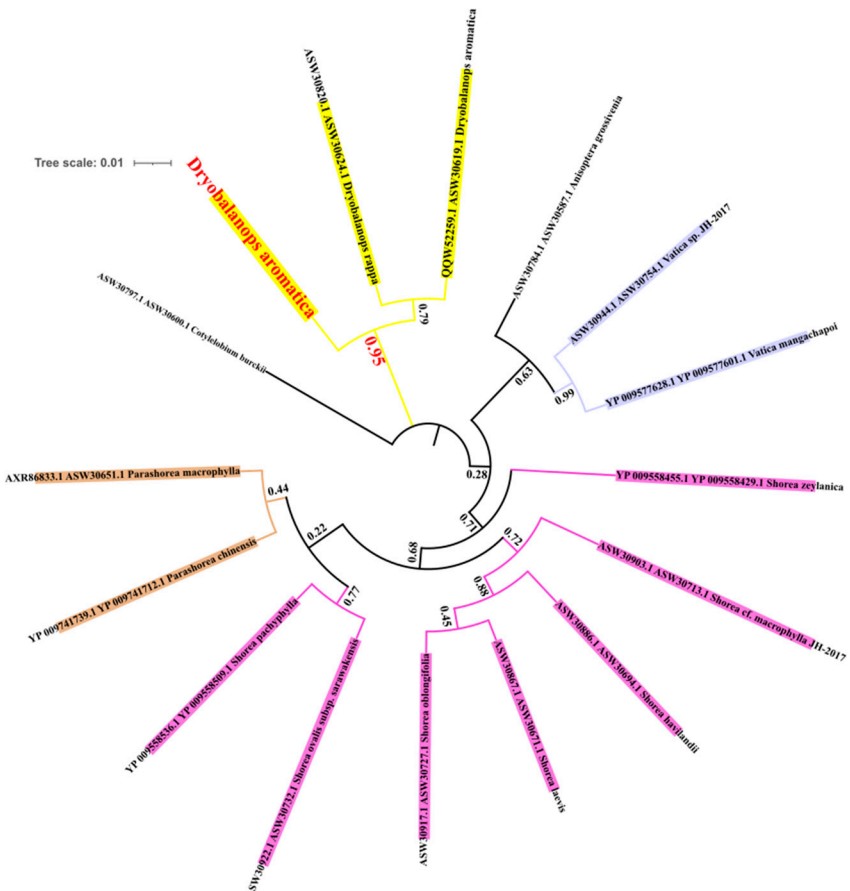

**Figure 6.** Phylogenetic tree of *D. aromatica* based on the combination of *mat*K and *rbc*L gene sequences.

## 4. Discussion

### 4.1. Genome Sequencing and Assembly

In this study, we present the genome sequencing of *D. aromatica* based on single sequencing run, which generated considerably higher output, 1,617,953,241 bases in 418,943 reads, compared with the output from the previous sequencing with the *Diospyros celebica* genome using MinION (318,492,700 bases in 302,567 reads) [63]. As much as 16.2% reads presented a quality score of 15 or higher with a mean quality score of 12.6. Raw read filtering yielded high-quality 1,559,878,347 reads with 15.7% presenting a quality score of 15 or higher and 100% of the total reads having a quality score of more than 7. Quality checks before and after screening were performed to determine information reads that will be gained in constructing contigs, scaffolds, or genome reconstructions.

Chloroplast genome sequencing of *D. aromatica* was successfully carried out with the aid of *D. turbinatus* as reference for assembly, which resulted in 148,856 bp of *D. aromatica* cpDNA. Long-read sequencing technologies, such as ONT, have the potential to allow a single read to cover the entirety or parts of the chloroplast genome, especially using the R9.4 chemistry [64,65]. In addition, ONT provides polishing tools (Medaka) to improve the accuracy of assembly results [66]. Despite concerns about the error rate exhibited by long-read sequencing technologies, genomic application to genetic resource conservation and plant breeding especially in forest trees can harness the benefit of these technologies [67].

### 4.2. Organization of a Partial Chloroplast Genome

The advantage of chloroplast DNA (cpDNA) is its small genome and stable structure. The low average nucleotide substitution makes the genome in cpDNA more conservative. In addition, the genome in cpDNA is not subject to recombination and is inherited uniparentally. In general, the genome size of a plant is approximately 61 Mb (*Genlisa tuberosa*) [66] to 152 Gb (*Paris japonica)* [68,69]. The chloroplast genome in terrestrial plants has been obtained with a size range of 120–170 Kb, although the chloroplast size of each plant is different [70]. For example, some studies showed a chloroplast genome size of 156 Kb (*Saurauia tristyla*) to 206 Kb (*Rhododendron griersonianum*) [51]. A study of the chloroplast genome of *D. turbinatus* (NC_046842.1), which is used as a reference for *D. aromatica* [32], successfully generated 5.2 Gb high-quality clean reads to assemble and obtain the annotation result as a complete chloroplast genome with a length of 152,279 bp.

Wang et al. [71] have recently determined that the complete chloroplast genome of *D. aromatica* is 152,696 bp in length. The two inverted repeats separated the two single-copy regions of 93,610 bp and 18,902 bp. A total of 128 predicted genes consisting of 84 protein-coding genes, 36 tRNA genes, and 8 rRNA genes were found. The GC content of the chloroplast genome was 39.16%. However, the result of our study could not be compared with those of Wang et al. [71] because the taxonomic classification of *D. aromatica* was inconsistent and incorrect, i.e., *D. aromatica* was designated as a species of the family Lauraceae. Differences in family groupings *of D. aromatica* between this study and Wang et al. [71] raised doubts on whether these two studies examined the same species. In addition, *D. aromatica* was clearly placed in the Dipterocarpaceae family [72–74] and not in the Lauraceae family [71].

The information of genes in the chloroplast genome will be useful to support the study of the evolutionary history of plants [75,76] and to select the best marker for *D. aromatica* in this case. Recent studies have shown that repeated chloroplasts serve a function that is useful in genetic resources for population genetics and biogeography studies [77]. In addition, other genetic studies in *Lycoris* species [78], *Acer miaotaiense* using *mat*K [79], and C4 plants using *rbc*L [80] reported that these genes could be used in for comparative and phylogenetic analysis. Some studies have proven that *mat*K and *rbc*L can be used as markers for plant evolutionary studies [79,81].

*4.3. Phylogenetic Inference*

The bootstrap values of *rbc*L (61%) and *mat*K (65%) in this study showed that the nodes were still allowed in the tree (above 50%) despite the low confidence [82], whereas the combined *rbc*L and *mat*K genes (95%) showed a very significant difference between *D. aromatica* and *D. rappa*. This pattern indicated that the combination of *mat*K and *rbc*L genes produced a better resolution of phylogenetic analysis [83]. The barcodes used for plant species are very unstable; thus, the more DNA markers used in analyzing a plant, the greater the effort to identify the plant species [84].

The data from this information provide new knowledge related to phylogenetic study on the genus *Dryobalanops* [85], which previously showed a phylogenetic relationship with the genera *Shorea* and *Hopea* [86]. In a previous study, the genus *Hopea* was declared as closely related to *Shorea* [87]. In a 1999 study, the paraphyletic group between *Hopea* and *Shorea* was revealed based on the *rbc*L sequence, leading to further questions of their phylogenetic relationship [88]. The *Dryobalanops* phylogeny was not resolved using the *rbc*L marker; however, the use of *trn*L-*trn*F placed *Dryobalanops* as a sister taxon to *Hopea* and *Shorea*. In another study in 2005 [89], *Dryobalanops* was considered as a sister taxon to a group containing *Shorea* and *Hopea*, but *Dryobalanops* is more closely related to *Shorea* than to *Hopea.* The next study in 2006 [90], in which a molecular phylogeny of the Indonesian Dipterocarpaceae was constructed using PCR-RFLP of the chloroplast regions *rbc*L, *pet*B, *psb*A, *psa*A, and *trn*L-F, revealed that *Dryobalanops* is intermediate between genera *Shorea* and *Dipterocarps.* However, the phylogenetic tree of *Dryobalanops*, especially *D. aromatica*, has not been studied extensively with other markers. The phylogenetic tree created from this study with *rbc*L marker shows that *Dryobalanops* is still included in the clade genus *Shorea,* but the genus *Vatica* was also included in one clade between *Shorea* and *Dryobalanops.* The *mat*K marker and combination of *rbc*L and *mat*K showed that *Dryobalanops* is monophyletic.

The bootstrap value showed the value of genetic relationships between species that have many similar characters and are closely related [91]. *D. rappa* and *D. aromatica* possibly descended from a common ancestor that carries the same chemical-genetic traits or patterns, which agrees with several studies that characterized the content of compounds in both species [92,93]. Several other studies have shown that *D. rappa* is an endemic species in Kalimantan with *D. beccarii*, *Dryobalanops fusca*, *Dryobalanops keithii*, and *D. lanceolata*, whereas *D. aromatica* and *Dryobalanops oblongiofolia* are endemic species from Sumatra [94]. The high proximity of *D. aromatica* and *D. rappa* means that the distribution of *D. aromatica* in Sumatra is an evolutionary form of Borneo, as evidenced by the fact that *D. aromatica* is found naturally in Kalimantan [95].

## 5. Conclusions

Genome sequencing of *D. aromatica* was successfully carried out by reading the base sequence of long-read DNA, resulting in high-quality DNA of 1.55 Gb from which a partial genome of *D. aromatica* chloroplasts of 148.856 bp was constructed. The processed data could be used to observe the genetic relationship of *D. aromatica* using two genes, namely, *mat*K, *rbc*L, and a combination of both. The phylogenetic tree showed that *D. aromatica* was closely related to *D. rappa* based on *mat*K gene markers and combinations (*mat*K and *rbc*L). However, the combination of *mat*K and *rbc*L genes showed a very high confidence level, so the combination of these genes is recommended for further analysis of *D. aromatica*.

**Supplementary Materials:** The following are available online at https://www.mdpi.com/article/10.3390/f12111515/s1, Table S1: Gene sequence from *rbc*L and *mat*K marker; Table S2: Protein-coding genes containing intron.



**Author Contributions:** Conceptualization, supervision and methodology, I.Z.S. and R.P.; Data curation and writing-original draft preparation, D.W.; software and formal analysis, R.P. and D.W.; Validation, R.P.; Resources, M.M., H.H.R., F.G.D. and R.P.; writing—review and editing, F.G.D., R.P. and H.H.R.; project administration, F.G.D. All authors have read and agreed to the published version of the manuscript.

**Funding:** The study was supported by the Ministry of Research and Technology/National Agency for Research and Innovation (RISTEK/BRIN) of the Republic of Indonesia for basic research scheme (Skema Penelitian Dasar) entitled "Pilot Sequencing of 100 Native Forest Tree Genomes to Support Ecosystem Restoration (Rintisan Sekuensing 100 Genom Pohon Hutan Asli untuk Mendukung Restorasi Ekosistem)", with contract No. 1/E1/KP.PTNBH/2021 between RISTEK/BRIN and IPB University and contract No: 2029/IT3.L1/PN/2021 between LPPM IPB University and Principal Investigator (Iskandar Z. Siregar).

**Acknowledgments:** The authors thank Laboratory of Forest Genetics and Molecular Forestry, Department of Silviculture, Faculty of Forestry and Environment, IPB University and Molecular Science Research Group in the Advanced Research Laboratory (ARLab), IPB University for facilitating this study.

**Conflicts of Interest:** The authors declare no conflict of interest.

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
