# Peer review of "Chloroplast Genome Draft of Dryobalanops aromatica Generated Using Oxford Nanopore Technology and Its Potential Application for Phylogenetic Study"

_forests, doi:10.3390/f12111515_

Round 1

Reviewer 1 Report

The title does not correspond to the content of the manuscript. A more correct version is “Analysis of genetic relationships for Dryobalanops aromatica using chloroplast genes”.

Introduction:

1) there is no information about the chloroplast genome and genes used for analysis;

2) the necessity of constructing the chloroplast genome and assessing the relationship for D. aromatica should be explained;

3) MinION Technology is described in great detail, but how its use will help to support conservation, breeding, and ecosystem restoration? Clarify please.

Results: It would be desirable to compare the results of sequencing the D. aromatica genome using MinION Technology and other NGS technologies.

Discussion:

1) the results from subsection 3.1 are not discussed;

2) has the relationship of D. aromatica with other species been previously assessed, for example, using molecular markers? If so, then you should compare your data with previously published ones;

3) the complete chloroplast genome of D. aromatica was recently published (Wang et al., Mitochondrial DNA B Resour., 2021, 6: 1687-1688). You should compare your data with previously obtained ones.

Conclusion:

L.294-296. “This processed data could be used as a showcase to observe the genetic relationship of D. aromatica using three markers, namely matK, rbcL, and a combination of both”. Only two markers were used for analysis, matK and rbcL, but the combination of markers is not the third marker. Correct please.

References:

Ref. 41 should preferably be replaced with a more authoritative source such as CBOL Plant Working Group, PNAS, 2009, 106: 12794-12797.

Ref. 59 is missing in the text, however, this is not a big loss for the manuscript.

Author Response

Dear Respected Reviewer,

We are grateful for the constructive comments from the reviewr#1.

Our responses including two three files and see the attachments.

Looking forward to your kind advices

Respectfully

Iskandar Z Siregar

Reviewer 2 Report

This research could be a valuable contribution to the field of genomic resources for dipterocarps.

Unfortunately, the text is hard to read because of the use of sometimes strange words and a really weird word order. Furthermore, it isn’t really clear to me what the authors wanted to show:

Just a proof that the sequencing method of MinIon is suitable for tropic species? That is what the title inclined. That wouldn’t be enough for a paper.

Or is the idea to show the annotated chloroplast genome? This genome is available since January 2021 in NCBI but without a belonging publication. So, is this the intention here? If it is this, this chapter is much too short. There is a very short chapter 3.2 „Chloroplast genome annotation“ where only a few words and a figure are used to tell the reader that the complete cp genome has been annotated – clearly more effort is given here to reduce all this to only two barcoding regions (rbcL and matK).

Or is it just to show a phylogeography using the barcoding sequences rbcL and matK? That it is what the authors are doing here, so far that I understood it. But, for a phylogeography with the use of only two barcoding markers a MinIon sequencing wouldn’t have been necessary. Thus, I do not really understand why the authors used MinIon technique when they just want to use two barcoding regions for a phylogenetic tree.

So, I recommend that the authors look for a clear common thread throughout the manuscript, choose an appropriate title and decide whether they would like to publish the results of MinIon sequencing (then they have to improve these parts intensively) or the phylogeography of the dipterocarps with the two barcoding markers.

Author Response

Dear Reviewer,

Thank you for your constructive comments and our apology for late responses.

Our responses are as attached. Looking forward to your additional constructive comments.

Respectfully
Iskandar Z Siregar

Round 2

Reviewer 1 Report

The manuscript by Wahyuni et al. has been improved. However, minor changes are still needed:

1) The relationship of D. aromatica with other species has previously been studied using molecular markers, for example, Yulita et al. (2005) Plant Species Biol. 20:167-182; Indrioko et al. (2006) Plant Syst. Evol. 261:99-115; Harnelly (2013) PhD Dissertation, Göttingen University. You should compare your data with previously obtained ones.

2) It is desirable to transfer Table 1 from the Materials and Methods to Supplementary Information.

Author Response

Dear Reviewer 1,

Thank you for your constructive comments and we have made revision accordingly as per your request. In addition, we have also done English Proofread by ENAGO service.

Look forward to your response.

Best wishes
Iskandar Z Siregar

Reviewer 2 Report

The authors did a lot of changes. The introduction is much better and more focused. The new title fits better.

But, the abstract hasn’t been changed so that it fit to the overall changed aims. The result section is only a little bit revised regarding some spelling/grammar errors. One of my concerns regarding the presentation of the chloroplast genome with only a few sentences in chapter 3.2 hasn’t improved. The authors mentioned in the response to me that there is a problem about the taxonomic determination and geographical distribution of D. aromatica and that they tried to discuss with the author of the published chloroplast genome. But I would like to read exactly this in the discussion. My comments are not meant to only give an answer to me but to improve the manuscript.

Unfortunately, the same is true for my next comment. The authors answered me about what they intend to do with the data but didn’t changed this part at the end of the introduction.

The authors rewrote sections of the discussion so the focus isn’t any longer on the method of MinIon but more on the phylogeny. And the language in the introduction and discussion is better. Thanks for this.

But still an overall reading of the text by a native speaker would be helpful.

Minor comment:

I have doubts, that a tree that can be grow up to 60 m will have a stem diameter of only 35-45 cm (line 43).

I guess that Table 1 (sequences of rbcL and matK) was added due to another reviewers comment? These are publically available sequences, thus I do not think that this must be presented in a Table in this publication.

Author Response

Dear Reviewer 2,

Thank you for your constructive comments and we have made revision accordingly as per your request. In addition, we have also done English Proofread by ENAGO service.

Look forward to your response.

Best wishes
Iskandar Z Siregar
